# The Crosstalk Analysis between mPSCs and Panc1 Cells Identifies CCN1 as a Positive Regulator of Gemcitabine Sensitivity in Pancreatic Cancer Cells

**DOI:** 10.3390/ijms25179369

**Published:** 2024-08-29

**Authors:** Beate Gündel, Xinyuan Liu, Anna Pfützenreuter, Veronika Engelsberger, Ralf Weiskirchen, J.-Matthias Löhr, Rainer Heuchel

**Affiliations:** 1Pancreas Cancer Research Lab, Department of Clinical Science, Intervention and Technology (CLINTEC), Karolinska Institutet, SE 141 86 Huddinge, Sweden; beate.gundel@ki.se (B.G.); matthias.lohr@ki.se (J.-M.L.); 2Institute of Molecular Pathobiochemistry, Experimental Gene Therapy and Clinical Chemistry (IFMPEGKC), RWTH University Hospital Aachen, D-52074 Aachen, Germany; rweiskirchen@ukaachen.de

**Keywords:** pancreatic ductal adenocarcinoma, PDAC, 3D cell culture, tumor–stroma crosstalk, drug resistance, co-culture

## Abstract

Pancreatic ductal adenocarcinoma (PDAC) is a deadly disease that is almost entirely resistant to conventional chemotherapy and radiation therapy. A significant factor in this resistance appears to be the dense desmoplastic stroma, which contains various cancer-associated fibroblast (CAF) populations. However, our understanding of the communication between tumor cells and CAFs that contributes to this aggressive malignancy is still developing. Recently, we used an advanced three-dimensional heterospecies, heterospheroid co-culture model to investigate the signaling between human pancreatic tumor Panc1 cells and mouse pancreatic stellate cells (mPSCs) through global expression profiling. Upon discovering that *CCN1* was significantly upregulated in Panc1 cells during co-culture, we decided to explore the role of CCN1 using CRISPR-Cas9 knockout technology. Panc1 cells lacking CCN1 showed reduced differentiation and decreased sensitivity to gemcitabine, primarily due to lower expression of genes involved in gemcitabine transport and metabolism. Additionally, we observed that stimulation with TGF-β1 and lysophosphatidic acid increased *CCN1* expression in Panc1 cells and induced a shift in mPSCs towards a more myofibroblastic CAF-like phenotype.

## 1. Introduction

Pancreatic ductal adenocarcinoma (PDAC) is one of the most lethal malignancies with a 5-year survival rate of less than 11% [1]. This low survival rate is largely due to late diagnosis and the lack of effective therapies. Interactions between pancreatic cancer cells and cancer-associated fibroblasts (CAFs), the predominant cell type in the desmoplastic stroma of PDAC, play a significant role in tumor initiation, progression, metastasis, and chemoresistance [2,3]. Therefore, therapeutic strategies targeting CAFs may hold promise for improving PDAC treatment outcomes. However, recent studies have shown conflicting results following CAF depletion, with some studies suggesting it may prevent disease progression while others indicate it could accelerate it [4,5,6,7]. Furthermore, there is evidence of functional heterogeneity within the CAF population in the PDAC stroma, highlighting the complex relationship between cancer cells and CAFs [8]. Consequently, it is crucial to deepen our understanding of how distinct CAF subtypes impact tumor biology in order to develop more effective treatment approaches for PDAC.

Recent studies have revealed several signaling pathways that differentially induce distinct subsets of CAFs. Öhlund et al. identified two functionally different CAF subtypes that originated from pancreatic stellate cells, known as myofibroblastic CAFs (myCAFs) and inflammatory CAFs (iCAFs) [8]. myCAFs have elevated expression levels of α-smooth muscle actin (α-SMA), are located in close proximity to tumor cells, and appear to encapsulate and locally restrict the tumor cells [8]. In contrast, iCAFs are located further away from tumor cells and lack α-SMA and express high levels of the chemokines interleukin 6 (IL6) and C-X-C motif chemokine ligand 1 (CXCL1), which support tumor progression [8]. However, the detailed interactions between CAF populations and neoplastic cells are still under investigation.

Cellular communication network factor 1 (CCN1), previously known as cysteine-rich angiogenic inducer 61 (CYR61), is a member of the CCN protein family. This family also includes connective tissue growth factor (CTGF/CCN2), nephroblastoma overexpressed (NOV/CCN3) gene, WNT1-induced secreted protein 1 (WISP-1) (CCN4), WISP-2 (CCN5), and WISP-3 (CCN6) [9,10]. CCN1 is classified as an immediate early gene and can be transcriptionally activated by various growth factors, including transforming growth factor-β1 (TGF-β1), fibroblast growth factor 2 (FGF2), and platelet-derived growth factor (PDGF) [11]. Additionally, agonists of G protein-coupled receptors (GPCRs) like lysophosphatidic acid (LPA), thrombin, and sphingosine-1-phosphate (S1P), as well as stress stimuli, can also induce CCN1 expression [11,12]. Due to a secretory signal peptide in exon-I, CCN1 is secreted into the extracellular space. There, it interacts with multiple components of the extracellular microenvironment due to its multimodal structure, containing an insulin-like growth factor-binding protein (IGFBP) homology domain, a von Willebrand factor type C repeat domain (vWC), a thrombospondin type 1 repeat (TSP1) and a C-terminal cysteine-knot (CK) domain. Accumulated data suggest that these domains may function as a hub integrating different integrins, heparan sulfate proteoglycan (HSPG) receptors, cytokines, and growth factor signaling pathways in a cell-type and context-dependent manner [11,13,14,15,16]. CCN1 targets various cell types including epithelial cells, fibroblasts, endothelial cells, smooth muscle cells, and neurons [14]. CCN1 plays crucial roles in both physiological processes, like embryonic development, senescence, tissue injury repair, and angiogenesis, as well as in pathological processes such as fibrosis and cancer [17,18,19]. Its functions in cellular processes are diverse and sometimes contradictory [13]. For instance, CCN1 can enhance cell survival and induce apoptosis, promote cell proliferation and trigger cell-cycle arrest, and support tumor growth while also suppressing tumorigenesis in different contexts [13,14].

Increased expression levels of CCN1 have been reported in pancreatic cancer and its metastatic lesions [20,21]. Additionally, CCN1 plays important roles in regulating epithelial to mesenchymal transition (EMT), stemness, neovascularization, and gemcitabine sensitivity in PDAC [22,23,24]. However, the regulatory mechanisms of CCN1, especially how it regulates the crosstalk between cancer cells and their microenvironments in PDAC, remain largely unknown.

In a recent work, where we investigated the cancer/stromal cell crosstalk in a 3D cell culture model, the CCN1 gene appeared as robustly upregulated in the cancer cell Panc1 when co-cultured with pancreatic stellate cells (PSCs) [25]. Here, we used the same model to study the direct communication between pancreatic cancer cells and PSCs, with a focus on the function and regulation of CCN1. To this end we knocked out CCN1 in the pancreatic cancer cell line Panc1 via the CRISPR-Cas9 technique. This resulted in heightened stemness and increased resistance to gemcitabine treatment. The interaction between neoplastic cells and PSCs, presumably through TGF-β1 and LPA, amplified CCN1 expression in Panc1 cells and affected the CAF subtype of PSCs.

## 2. Results

### 2.1. The Expression of CCN1 in Panc1 Cells Is Affected by Co-Culture with mPSCs

In our previous RNA-seq profiling [25], the expression level of *CCN1* was significantly higher in Panc1 cells from heterospheroids (co-cultured with mPSCs) compared to Panc1 cells from monospheroids (Figure 1A), as confirmed with RT-qPCR (Figure 1A). According to the RNA-seq profiling, there was no difference in *Ccn1* expression in mPSCs from monospheroids and heterospheroids (Figure 1B). However, upon performing validating RT-PCR, we found a decreased expression of *Ccn1* in mPSCs from heterospheroids compared to monospheroids (Figure 1B). To investigate the functions of CCN1 in pancreatic cancer, we knocked out *CCN1* in Panc1 cells using transient plasmid-based CRISPR-Cas9 technology. We identified clones C3, C7, and F3 as Panc1-CCN1-KO via Western blot (Figure 1C) and Sanger sequencing (Appendix A). The morphologies of Panc1-WT and Panc1-CCN1-KO monospheroids at different time points are depicted in Figure 1D. While Panc1-WT and Panc1-CCN1-KO F3 formed more compact spheroids from day 4 onwards, Panc1-CCN1-KO C3 and C7 monospheroids showed a looser structure throughout, suggesting impaired cell–cell connectivity.

### 2.2. CCN1 Negatively Regulates Stemness Genes and Positively Affects Gemcitabine Transporter and Metabolizing Genes

The mRNA expression of genes affected by CCN1 was assessed in Panc1-CCN1-KO C3, C7, and F3 clones from monospheroids and heterospheroids and compared to the mRNA expression of Panc1-WT. The expression of the stemness marker *CD24* was significantly higher, whereas the epithelial markers *CK19* and *CDH1* were lower in Panc1-CCN1-KO cells in monospheroids (Figure 2A) and heterospheroids (Appendix A), indicating that CCN1 may negatively regulate stemness in Panc1. Compared to Panc1-WT, Panc1-CCN1-KO cells had lower expression levels of the matricellular protein *CTGF*, the proliferation marker *MKI67* and the SHH signaling mediator *GLI1* in both monospheroids (Figure 2A) and heterospheroids (Appendix A). In our previous work, we had observed that the mRNA expression of the gemcitabine transporter (*SLC29A1*) and the gemcitabine activating enzyme (*DCK*) was increased in Panc1 cells when co-cultured in heterospheroids with mPSCs [25]. Interestingly, both genes showed lower mRNA expression in the Panc1-CCN1-KO cells in monospheroids (Figure 2A) and heterospheroids (Appendix A). Semi-quantitative Western blot analysis confirmed the significant decrease in DCK protein expression in Panc1-CCN1-KO cells (Figure 2B). To further support the correlation between *CCN1* and its related genes, we analyzed the publicly available microarray dataset GSE71729 of PDAC tumor samples. The mRNA expression of *CCN1* was significantly inversely correlated with *CD24* but positively correlated with *CTGF*, *GLI1*, and *DCK* expression in PDAC patient samples (Figure 2C), corroborating the results of our spheroid model. Direct correlation between CCN1 and eCK expression was further demonstrated by rescuing the CCN1-KO phenotype in the clone Panc1-CCN1-KO C7 (Figure 2D). Restoring CCN1 protein expression also restored DCK protein expression while the control protein (enhanced Green Fluorescent Protein, eGFP) caused no elevation of DCK. Taken together, these findings suggest that CCN1 may play a role in affecting cellular plasticity (stemness and epithelial phenotype), regulation of cell adhesion (CTGF), SHH signaling, and gemcitabine sensitivity.

### 2.3. Lack of CCN1 Decreases Chemotherapy Sensitivity for Gemcitabine but Not Paclitaxel and SN38

Since the KO experiments suggested that *CCN1* positively regulates *SLC29A1* and *DCK* expression, which may promote gemcitabine action, the chemosensitivity in Panc1-CCN1-KO cells towards gemcitabine was evaluated and compared to the chemosensitivity in Panc1-WT cells. Cells were grown both in classical monolayer and as monospheroids. The cultures were then treated on day 1 after seeding with different doses of gemcitabine, and cell viability was detected on day 4 via CellTiter-Glo^®^ 3D Cell Viability Assay. We found that the growth inhibition of gemcitabine was more effective in Panc1-WT compared to Panc1-CCN1-KO cells independent of 2D (Figure 3A, Appendix A) or 3D culture (Figure 3B, Appendix A). To investigate whether CCN1 affects gemcitabine-induced apoptosis, we analyzed the epithelial-specific ccCK18 in Panc1-WT and Panc1-CCN1-KO cells from both monospheroids and heterospheroids using the M30 Apoptosense^®^ CK18 kit. Results showed that Panc1-CCN1-KO monospheroids compared to Panc1-WT had less apoptosis, indicated by a lower ccCK18 ratio of treated versus untreated (Figure 3C). However, co-culturing with mPSCs significantly increased gemcitabine-induced apoptosis in Panc1-WT and Panc1-CCN1-KO cells from heterospheroids (Figure 3C). Interestingly, there was no significant difference between Panc1-WT and the Panc1-CCN1-KO clones for the ratio of ccCK18 heterospheroid versus monospheroids (Figure 3D), suggesting that mPSCs may compensate for the lack of Panc1-expressed CCN1 in the microenvironment of heterospheroids. Therefore, we analyzed the expression of *Ccn1* in mPSCs from heterospheroids and found that mPSCs had significantly increased *Ccn1* expression levels when co-cultured with Panc1-CCN1-KO cells compared to Panc1-WT cells (Figure 3E). To determine whether CCN1 might also influence the sensitivity to other chemotherapeutic drugs used in PDAC therapy, we treated Panc1-WT and Panc1-CCN1-KO monospheroids with paclitaxel or SN38 at a concentration of 1 μM. Although paclitaxel and SN38 treatment impaired the viability of all cells compared to NC, the relative cell viability did not significantly differ between WT and CCN1-KO clones and did not show a consistent trend within the different clones (Figure 3F,G). These data suggest that the lack of CCN1 has no significant influence on the action of paclitaxel and SN38 in Panc1 monospheroids, but instead suggests CCN1 is specifically involved in the metabolization and activation of gemcitabine.

### 2.4. LPA and TGF-β1 Upregulate CCN1 Expression in Panc1

LPA and TGF-β1 signaling have been reported to regulate CCN1 in breast cancer and prostate epithelial cells [26,27]. Our previous RNA-seq profiling results indicated that mPSCs in heterospheroids had increased *Tgfb1* and *Enpp2* (ectonucleotide pyrophosphatase/phosphodiesterase family member 2/autotaxin, a protein converting lysophosphatidylcholine into LPA) expression [25], which was further verified with RT-PCR (Appendix A). We therefore hypothesized that TGF-β1 and LPA might also be involved in the regulation of CCN1 expression in pancreatic cancer cells. To verify this, we stimulated Panc1-WT monospheroids with either 20 μM LPA or 5 μg/mL recombinant TGF-β1 on day 1 after seeding under low-serum conditions, to minimize the effects of the phospholipids and TGF-β1 contained in FCS. As expected, stimulation with LPA significantly increased the expression of *CCN1* and its downstream genes *CTGF*/*CCN2* and *ITGA5*, as well as the proliferation marker *MKI67*. However, it only marginally affected the expression of *SLC29A1*, *DCK*, and *TGFB1* (Figure 4A). TGF-β1 stimulation also significantly increased the expression of *CTGF*/*CCN2* and *ITGA5,* as well as *SLC29A1*, but it did not affect *CCN1*, *MKI67*, and *DCK* expression in a statistically significant manner (Figure 4A). However, when we stimulated Panc1-WT monospheroids with TGF-β1 under a high-serum condition, the increase in FCS led to an elevated expression of *CTGF*/*CCN2* and *ITGA5* as well as *CCN1* (Figure 4B).

To further confirm the effect of LPA and TGF-β1 on *CCN1* expression, we blocked Enpp2-related LPA generation and TGF-β1 pathways using the autotaxin/ENPP2 inhibitor PF8380 [10 μM] and the selective TGF-β receptor type I/II inhibitor (TGFBRi) LY2109761 [5 ng/mL] in Panc1 heterospheroids grown under high-serum conditions on day 5. This timepoint was chosen as increased *CCN1* expression was detected in Panc1 from heterospheroids compared to monospheroids (Figure 1A,B). The expression of *CCN1* was slightly reduced after incubation with either PF8380 or LY2109761 compared to the negative control (Figure 4C). However, the combination of both inhibitors resulted in a strong and statistically significant reduction in *CCN1* expression in Panc1-WT cells from heterospheroids (Figure 4C). The expression of *DCK* was not affected by single incubation with PF8380 or LY2109761, nor by the combination of both inhibitors (Figure 4C). Notably, the expression of *SLC29A1* was significantly decreased by LY2109761 alone, and the combination with PF8380 did not result in any further decrease (Figure 4C). Additionally, the analysis of the publicly available microarray dataset GSE71729 confirmed a positive correlation between *CCN1* and *ENPP2*, as well as *TGFB1* expression in pancreatic cancer patients (Figure 4D). We also aimed to determine the possible source(s) of LPA and TGF-β1 that upregulated *CCN1* expression in Panc1/mPSCs heterospheroids. By comparing the expression of *ENPP2*/*Enpp2* and *TGFB1*/*Tgfb1* in Panc1 and mPSCs between heterospheroids and monospheroids in our previous RNA-seq profiling [25], we found that the expression of *Enpp2* was almost exclusively contributed by mPSCs (Appendix A), since we could not detect ENPP2 expression in Panc1 cells. This is because *ENPP2* expression is not detectable in Panc1 via RNA-seq and RT-PCR. While both Panc1 and mPSC monospheroids expressed *TGFB1*/*Tgfb1*, the expression of *Tgfb1* was further increased in mPSCs from heterospheroids compared to mPSC monospheroids. There was no difference in *TGFB1* expression in Panc1 cells from heterospheroids and monospheroids (Appendix A). Therefore, the increased expression of *Tgfb1* and *Enpp2* from mPSCs might explain the elevated expression level of *CCN1* in Panc1 cells from heterospheroids.

### 2.5. LPA and TGF-β1 Stimulate Ccn1 Expression in mPSCs and Enhance the myCAF Subtype

To further investigate the impact of downstream signaling from LPA and TGF-β1 on mPSCs, we treated our mPSC monospheroid cultures with 20 μM LPA or 5 μg/mL recombinant human TGF-β1. TGF-β1 increased the expression of *Ccn1* and the myCAF markers *Acta2*, *Ctgf*, and *Col1a1*, while decreasing the iCAF marker *Cxcl1* in mPSCs (Figure 5A). Furthermore, stimulation with LPA also led to a significant increase in *Ccn1* expression and a decrease in *Enpp2* expression in mPSCs (Figure 5B). LPA treatment did not impact *Tgfb1* expression but did result in a notable increase in the myCAF markers *Acta2* and *Ctgf*, along with a slight decrease in the iCAF marker *Cxcl1* (Figure 5B), suggesting that LPA may play a role in influencing CAF subtypes. To confirm this, we exposed mPSCs/Panc1-WT heterospheroids grown under high-serum conditions on day 5 to the autotaxin inhibitor PF8380 for 72 h. Interestingly, the inhibitor significantly reduced the expression of the myCAF markers *Acta2* and *Ctgf* (Figure 5C). Additionally, analysis of the publicly available microarray dataset GSE71729 from bulk tissue of PDAC patients revealed a positive correlation between autotaxin and *ACTA2* expression (Figure 5D). Overall, these findings indicate that LPA and TGF-β1 signaling increase *Ccn1* expression in mPSCs and promote a more myCAF-like phenotype in mPSCs.

## 3. Discussion

The extensive stromal reaction, especially due to the heterogeneous fibroblast subtypes, plays important roles in PDAC [28]. We aimed to better understand the tumor–stroma crosstalk, with the hope of finding vulnerabilities that can be exploited in the search for more effective therapies for this devastating disease. PSCs/CAFs are a key component of the stroma, which play critical roles in gemcitabine resistance; however, ablating CAFs via pharmacologic therapy in clinical trials showed disappointing results [28]. Instead, therapy strategies should aim at normalizing the stromal cells or interfering with the tumor–stromal crosstalk. We found that mPSCs increased gemcitabine sensitivity of the Panc1 tumor cell line upon direct 3D co-culture (Figure 3C) [25]. In addition, the presence of mPSCs also resulted in higher levels of expression for *CCN1* in Panc1 cells (Figure 1A). Notably, *CCN1* knockout in Panc1 cells increased the resistance towards gemcitabine treatment (Figure 3A–C, Appendix A). Moreover, signaling via TGF-β1 and LPA between Panc1 and mPSCs upregulated *CCN1* expression and altered mPSCs towards a more myCAF-like phenotype (Figure 4 and Figure 5).

To understand the signals orchestrating tumor–stromal interactions, various in vivo and in vitro models have been utilized. The traditional cell monolayer culture is the most commonly used model, but it does not accurately represent the microenvironment of 3D cell interactions, limiting the ability to study tumor–stroma crosstalk [25,29]. Genetically engineered mouse models (GEMMs) have been shown to effectively model PDAC progression, but they are costly and time-consuming to maintain. Additionally, it is challenging to uncover interactions between different cell types in this highly complex tumor microenvironment [29]. In order to find a meaningful compromise between traditional monolayer culture and GEMMs, we developed an advanced three-dimensional (3D) heterospecies, heterospheroid model. This model involves co-culturing tumor cells and PSCs, allowing for the study of tumor–stroma crosstalk and genetic manipulation [25,30]. We have demonstrated that the heterospecies, heterospheroid model exhibits key characteristics of PDAC and enables direct detection of gene expression in tumor cells and PSC separately. This contributes to detailed investigations of cell signaling networks [25,30].

In this work, we validated our previous finding that the expression of *CCN1* in Panc1 was increased by co-culturing with mPSCs (Figure 1A) [25]. However, the expression of *Ccn1* in mPSCs was lower in heterospheroids with WT-Panc1 cells compared to mPSC monospheroids (Figure 1B), suggesting a tight regulation of CCN1/Ccn1 in the extracellular space. This idea was supported by the finding that the *Ccn1* expression in mPSCs was significantly increased when co-cultured with Panc1-CCN1-KO instead of Panc1-WT cells (Figure 3E), presumably in an attempt to compensate for the lack of Panc1-contributed CCN1. CCN1 seems to positively affect cellular plasticity, since CRISPR-Cas9 knockout of *CCN1* in Panc1 resulted in increased expression of the stemness marker *CD24* and decreased expression of epithelial markers *CK19* and *CDH1* (Figure 2A), representing a more dedifferentiated state. Cellular plasticity has been recognized as a new hallmark of cancer, influencing malignant progression and therapy resistance [31]. Accordingly, our study showed that the more dedifferentiated Panc1-CCN1-KO cells were more resistant to gemcitabine (Figure 3A–C, Appendix A), which can be partly explained by the reduced mRNA expression of the gemcitabine importer *SLC29A1* and the gemcitabine activating enzyme *DCK* at both the mRNA and protein levels (Figure 2A,B). The correlation of DCK and CCN1 expression was further validated by restoring CCN1 expression in Panc1-CCN1-KO cells which also restored DCK expression to wild-type levels (Figure 2D). These findings are also supported by the observation that gemcitabine sensitivity increased in Panc1 cells grown in heterospheroids compared to monospheroids (Figure 3C), as paracrine CCN1 signaling by PSCs compensated for the lack of CCN1 protein in Panc1-CCN1-KO cells. Of note, the lower apoptosis level in Panc1-CCN1-KO cells is brought up to the Panc1-WT level via co-culture with mPSCs in heterospheroids (Figure 3C,D), possibly through upregulation of *Ccn1* in the co-cultured mPSCs (Figure 3E). The public microarray dataset from PDAC patients also supported that *CCN1* was negatively correlated with *CD24*, but positively correlated with *DCK* (Figure 2C). A previous study demonstrated that high levels of SLC29A1 and DCK are related to longer survival times for PDAC patients with adjuvant gemcitabine therapy [32]. Therefore, our results imply that *CCN1* might act as a molecular mechanism that counteracts stemness and gemcitabine resistance in PDAC patients. However, others suggested that *CCN1* might promote EMT and stemness [23] and increase gemcitabine resistance via downregulation of DCK and SLC29A1 in PDAC [20,24]. Therefore, the combination therapy of CCN1 inhibition with gemcitabine was considered a promising strategy [33]. The function of CCN1 in these studies has been deduced from sh-RNA knock down [24] and CRISPR-Cas knock out [20] studies in Panc1 cells. However, a significant difference between those and our study lies in their approach of genetic manipulation of the *CCN1* gene which used lentiviral-based shRNA expression or CRISPR-Cas9 editing [20,24]. RNAi technology was adopted firstly as a gene-silencing technique; however, the high off-target effects inducing unwanted phenotypes and incomplete silencing and knock-down instead of knock-out, challenging the results of such studies [34]. The CRISPR-Cas technique can produce permanent genetic modifications, blocking protein expression completely, which eliminates the confounding effects from remnant protein expression. The use of lentiviral expression in both approaches is very effective, but has the disadvantage of unknown genomic integration of the lentivirus risking disruption to gene expression via insertional mutagenesis [35], and in the case of the CRISPR approach, the constitutive expression of the editing enzyme Cas9 can cause additional off-target genomic editing. On the other hand, we used the lipofection method for gene-knockout via the CRISPR method, which is easy to conduct, economical, and with minimal toxicity due to only transient expression of Cas9. The CCN1-KO cell clones derived from single cells were verified via Western blot and Sanger sequencing before starting the investigation of gene function. In light of our results and those of others, conclusions about the exact roles of CCN1 in pancreatic cancer cells need further studies as other effectors might be involved too, which lead to seemingly contradicting results. This also implies that CCN1 inhibitor therapy would warrant extreme caution, and experiments should be carefully considered regarding the applied biological model.

By using Panc1-WT monospheroids, we discovered that the expression of *CCN1* and its downstream genes *CCN2*/*CTGF* and *ITGA5* was positively regulated by LPA and TGF-β1 (Figure 4A,B). In addition, we noticed that TGF-β1 increased *SLC29A1* expression, while the selective TGF-β receptor type I/II inhibitor LY2109761 decreased *SLC29A1* expression in Panc1-WT cells (Figure 4A,C). Further investigation is required to determine if the upregulation of *SLC29A1* after stimulation with TGF-β1 is due to the upregulation of *CCN1*. Although we noted a positive correlation between *SLC29A1* and *DCK* with *CCN1* (Figure 2A–C), and stimulation with LPA and TGF-β1 upregulated *CCN1* expression in Panc1-WT cells (Figure 4A), the expression of *DCK* in Panc1-WT cells was not influenced by LPA and TGF-β1 or their respective inhibitors (Figure 4A,C), suggesting a regulation of DCK independent of TGFB and LPA. LPA and the autotaxin inhibitor only slightly affected the expression of *SLC29A1*, despite robustly up- and downregulating *CCN1* expression (Figure 4A,C). These results suggest that the expression of *SLC29A1* is greatly impacted by CCN1, possibly through TGF-β1, while the expression of DCK seems to be influenced by CCN1 independently of LPA and TGF-β1 signaling.

A previous study has shown that PDAC cells activate neighboring PSCs to secrete abundant LPC, and express autotaxin/ENPP2, which converts the local LPC to LPA-signaling lipid [36]. Additionally, our work demonstrated that activated PSCs are the main source of autotaxin in our heterospheroid model since the expression of autotaxin in Panc1 is not detectable. We also observed negative feedback between LPA and Enpp2 in mPSC monospheroids (Figure 5B). Although the upregulation of CCN1 mRNA by TGF-β1 and LPA is not entirely new (Figure 4A) [26,27], our heterospecies, heterospheroids model highlighted an unanticipated role for PSCs in PSC-expressed *Tgfb1* and *Enpp2* (Appendix A) collaborating to induce *CCN1* expression in Panc1 cells. TGF-β1 is a well-established activating signal for fibroblasts [37,38] and has been reported to induce myCAFs formation [25,39]. In our spheroid model, we observed that not only TGF-β1 but also lipid-signaling autotaxin LPA could stimulate mPSCs to achieve a more myCAFs-like state with increased expression of the myCAFs marker *Acta2* but decreased expression of the iCAF marker *Cxcl* (Figure 5A,B).

Gemcitabine is the first-line chemotherapeutic agent for all stages of PDAC patients, but patients develop resistance within weeks after chemotherapy initiation [40]. There is abundant evidence that the dense fibrosis in the tumor microenvironment induces both intrinsic and extrinsic gemcitabine resistance [41,42,43,44,45]. In this regard, therapeutic strategies targeting CAFs were developed. The most debated one might be stromal-depletion via blocking the SHH pathway, which showed promising results in pre-clinical studies but failed in a clinical phase-II trial [6,46,47]. Further studies revealed that depletion of CAFs through pharmacological or genetical blockage of Shh signaling in mice resulted in more dedifferentiated, aggressive tumor cells [5,40,48]. Another study uncovered CAFs as a sink for gemcitabine [42]. Some light might be shed on these partly discordant results by the discovery of distinct fibroblast populations, including myCAFs, iCAFs, and apCAFs, but their exact functional roles and mechanisms through which they develop are still under investigation [8,39,49]. We have developed a co-culture model whereby PSCs shift towards myCAF or iCAF phenotypes upon co-culture with Panc1 under different culture conditions, especially under nutrient differences [25]. Previous studies had suggested that αSMA-positive CAFs acted to restrain tumor progression, most probably through production of type I collagen that mechanically restrained tumor spread [48,50,51]. By using this co-culture model, we now revealed the new function of myCAFs, that is, increasing gemcitabine sensitivity of Panc1 cells, presumably through upregulation of *CCN1* expression, adding another puzzle piece to the anti-tumor/protector function of PSCs. Considering the overall tumor-suppressive roles of myCAFs (αSMA-positive CAF) in PDAC, converting tumor-promoting iCAFs into tumor-restraining myCAF may be another promising treatment strategy.

In conclusion, our study suggests that the signaling of TGF-β1 and LPA between mPSCs and Panc1 cells shifts mPSCs to a more myCAF-like (tumor-suppressive) phenotype and increases the expression of *CCN1* in Panc1 cells. CCN1 in Panc1 cells appears to be involved in the regulation of cellular plasticity and chemosensitivity towards gemcitabine, two very important factors for the treatment of PDAC. Considering that CAF ablation therapy is still controversial, our results indicate a potential therapeutic opportunity in re-educating CAFs towards a tumor suppressing phenotype.

## 4. Materials and Methods

### 4.1. Cell Lines and Monolayer Cell Culture

The Panc1 cell line, a well-characterized human pancreatic cancer cell line, was purchased from the American Type Culture Collection (ATCC) [52]. Positive authentication of Panc1 cells was conducted by ATCC using Promega’s PowerPlex^®^ 18D system to determine short tandem repeat profiles. The immortalized mouse PSC clone 2 (imPSCc2), referred to as mouse PSC (mPSC), was generously donated by Dr. Raul Urrutia and Dr. Angela Mathison at the Mayo Clinic College of Medicine, Rochester, Minnesota [53]. Both Panc1 and mPSCs were cultured in DMEM (Gibco 31053044) medium supplemented with 10% fetal bovine serum (FBS) (Gibco 10270106), GlutaMAX™ Supplement (Gibco, Waltham, MA, USA, 35050061), and 0.5% penicillin/streptomycin (Gibco 15070063) at 37 °C in a 5% CO_2_ humidified environment [25]. The absence of mycoplasma was routinely checked using the MycoAlert^TM^ PLUS Mycoplasma Detection Kit (LT07-705, Lonza, Basel, Switzerland).

### 4.2. Generation of CCN1 Knockout Pancreatic Cancer Cell Line Using CRISPR/Cas9 Technique

Panc1 cells were cultured in DMEM medium supplemented with 10% FBS and penicillin/streptomycin. Equimolar mixtures of two synthetic single-guide RNAs (5′-AGCCCUGCGACCACACCAAG-3′ and 5′-CUGCGCCAAGCAGCUCAACG-3′, CRISPR evolution sgRNA EZ Kit, Synthego, Redwood City, CA, USA; 1 µg total) were used to target CCN1 in Panc1 cells. The cells were seeded at 4 × 10^5^ cells/well in a 6-well plate and transfected the next day with guide RNAs pre-complexed with 5 µg Alt-R^®^ Sp. Cas9 Nuclease V3 (IDT) using Lipofectamine™ CRISPRMAX™ Cas9 Transfection Reagent (ThermoFisher Scientific, Waltham, MA, USA) following the manufacturer’s instructions. The cells were expanded for several days before genomic DNA extraction (QIAamp mini kit, Qiagen, Germantown, MD, USA) was performed. The frequency of edited alleles was estimated using droplet digital PCR (QX200 System, BioRad, Hercules, CA, USA). Individual cells from the edited cell pools were sorted into 96-well plates with an SH800s cell sorter (Sony Biotechnology, San Jose, CA, USA). After expansion, protein extracts were generated for Western blotting.

### 4.3. Western Blotting

Panc1-WT and Panc1-CCN1-KO clones were lysed in T-PER Tissue Protein Extraction Reagent (Thermo Fisher, 78510) containing a protease inhibitor cocktail (Roche, Basel, Switzerland, 05892970001). After lysis, cell debris was removed via centrifugation (12,000× *g*). The protein concentration was quantified using the Bradford protein assay (Bio-Rad, 5000201). Volumes containing 30 µg of protein were used for polyacrylamide gel electrophoresis with sodium dodecyl sulfate (SDS-PAGE). Gradient polyacrylamide gels (Bio-Rad, 4561094) were utilized for optimal separation, followed by blotting to a nitrocellulose membrane (Bio-Rad, 1704158) using the Trans-Blot Turbo transfer system. Membranes were blocked for 2 h in 1X Tris-buffered saline with 0.1% Tween 20 buffer (TBS-T, ThermoFisher, #28360) containing 5% bovine serum albumin (BSA, Sigma-Aldrich, St. Louis, MO, USA, A7906) and then incubated with primary antibodies (pAB): CCN1 (Boster Bio, PB9549, 1:1000), DCK (LifeSpan BioSciences, Lynnwood, WA, USA, LS-B10837, 1:1000), and Tubulin (Abcam, Cambridge, UK, ab7291, 1:1000) overnight or for 2 days at 4 °C. This was followed by incubation with fluorophore-coupled secondary antibodies (sAB), goat-anti-rabbit-AlexaFluor488 (Invitrogen, Waltham, MA, USA, A11008, 1:10,000) and goat-anti-mouse-AlexaFluor647 (Invitrogen, A21235, 1:10,000), for 1 h at room temperature. Antibody binding was detected via the VersaDoc imaging system (BioRad) with exposure times ranging from 20 s to 60 s. For semi-quantitative Western blotting, protein signal quantification was performed using Image Studio Lite Ver 5.2. The amount of DCK per lane was divided by the tubulin content of the same lane to obtain a ratio. The ratio of DCK/Tub in WT Panc1 cells was set to 1, and the ratios in CCN1 knockout clones were normalized to Panc1-WT signals. Three independent biological replicates were conducted, and statistical analyses were performed using Student’s *t*-test for individual samples.

### 4.4. Mutation Analysis of Panc1-CCN1-KO Clones

Genomic DNA was extracted from Panc1 WT and three Panc1-CCN1-KO clones (C3, C7, and F3). Regions of interest were amplified from genomic DNA using KAPA HiFi Hot start (Roche) and specific oligos for CCN1 (FW 5′-GGACGAGATCAGAGGCTC-3′ and REV 5′-CAGACACACTGAATTGCATTC-3′) PCR products were gel purified (Freeze N’ Squeeze, BioRad) and sequenced (Eurofins Genomics, Louisville, KY, USA). Sequencing traces were deconvoluted with the DECODR software (version 3.0) [54].

### 4.5. Spheroid Cell Culture

Spheroid culture for Panc1-WT, Panc1-CCN1-KO C3, C7, and F3 clones, as well as mPSCs, was carried out as described in depth [25]. On day 5, spheroids were harvested for RNA isolation. Spheroid were imaged using an Olympus IX81 inverted microscope at a total magnification of 40×, with a scale bar of 500 μm [25].

### 4.6. TGF-β1 and LPA Stimulation Assay

Panc1-WT cells and mPSCs were grown as monospheroids in either high-serum (10% FCS) condition or low-serum condition. In the low-serum condition, DMEM medium was supplemented with 0.1% FBS, 0.5% bovine serum albumin fatty acid free (BSA-FAF, Sigma, A8806), 1/1000 insulin-transferrin-selenium-sodium pyruvate (ITS-A, Gibco^TM^, ThermoFisher), GlutaMAX™ supplement, and 0.5% penicillin/streptomycin, as well as 0.24% methyl cellulose. Panc1-WT and mPSC monospheroids were stimulated with either 5 ng/mL recombinant human TGF-β1 (PreproTech Nordic, Waltham, MA, USA, 100-21-10) [25] or 20 μM 18:1 lysophosphatidic acid/LPA (Avanti^®^ Polar Lipids, Inc., Alabaster, AL, USA, 857130P) on day 1 and harvested on day 4 after seeding for RNA isolation.

### 4.7. Inhibition of TGF-β1 and LPA Signaling

Panc1-WT/mPSC heterospheroids were established in high-serum conditions and treated with 5 μM TGF-β receptor kinase inhibitor LY2109761 (SML2051, Sigma-Aldrich) or 10 μM autotaxin (ATX) inhibitor PF8380 (SML0715, Sigma-Aldrich) on day 5, either individually or in combination. The heterospheroids were then collected on day 8 for RNA extraction.

### 4.8. Gene Expression Analysis via Quantitative Real-Time PCR (RT-qPCR)

Spheroids were collected in 15 mL tubes and centrifuged at 2000 rpm at 4 °C for 5 min. The supernatant was removed, and the spheroids were washed once with cold PBS before being centrifuged again. Subsequently, the PBS was discarded, and RNA extraction was carried out using the QIAshredder (Qiagen, 79656) and the RNeasy Mini Kit (Qiagen, 74104) following the manufacturer’s instructions. The concentration of RNA was measured using the NanoPhotometer^®^ NP80. Complementary DNA (cDNA) was synthesized by transcribing 250 ng of RNA with the iScript cDNA synthesis kit (Bio-Rad, 1708891) following the protocol for cultured cells. The design and validation of species-specific primers (Appendix A) was performed as previously described [25]. Primers were designed based on the sequence differences between mouse and human homologs using PRIMER3 (v.0.4.0) and were further verified by testing on cDNA from both human and murine cell lines. The RT-qPCR reaction was carried out as described previously using the Thermo Scientific™ Maxima SYBR Green/Fluorescein qPCR Master Mix kit (ThermoFisher Scientific, K0243) [25]. The house keeping genes *RPL13A*/*Rpl13a* for humans or mice were used to normalize target gene expression, and changes in gene expression levels were calculated using the 2^−ΔΔCt^ method. Three independent biological replicates were performed for each gene, and statistical analyses were conducted using Student’s *t*-test on individual samples. The common nomenclature of all capital letters for human genes and small letters with first capital letter for murine genes is used in this article.

### 4.9. Drug Testing and Cell Viability Assay

We cultured Panc1-WT cells and Panc1-CCN1-KO cells from clones C3 and C7 in traditional monolayer and as monospheroids with 2500 cells per well in 96-well plates under high-serum conditions to investigate the effect of CCN1 on the sensitivity of chemotherapeutic drugs. Different doses (1, 5, and 50 μM) of gemcitabine (Sigma-Aldrich, G6423), 1 µM paclitaxel (Sigma-Aldrich, 580555), and 1 µM SN38 (an active metabolite of irinotecan; Sigma-Aldrich, H0165) were added on day 1, and cell viability was assessed on day 4 using the ATP-based CellTiter-Glo^®^ 3D Cell viability assay (Promega, Madison, WI, USA, G9681) following the manufacturer’s instructions [25]. Luminescence was measured using a SpectraMax i3x microplate reader (Molecular Devices, San Jose, CA, USA). The relative cell viability was calculated by comparing the luminescence of cells treated with chemotherapeutic drugs to that of the respective negative control. Statistically significant differences between Panc1-WT and Panc1-CCN1-KO cells were determined using Student’s *t*-test (individual samples).

### 4.10. Epithelial-Specific Apoptosis Assay

Panc1-WT and Panc1-CCN1-KO monospheroids and heterospheroids (co-cultured with mPSCs) were grown under high-serum condition and treated with 50 μM gemcitabine on day 1 for 72 h. The M30 Apoptosense^®^ CK18 Kit (Diapharma, West Chester Township, OH, USA, #P10011) was used following the manufacturer’s instructions to quantitatively detect the epithelial-specific caspase-cleaved cytokeratin 18 (ccCK18) on day 4. The relative cell apoptosis rate was determined by normalizing the ccCK18 value for gemcitabine treated spheroids to the negative control, and differences between samples were assessed with Student’s *t*-test.

### 4.11. Microarray Dataset Analysis

The mRNA microarray expression data from 145 PDAC patients (GSE71729 [55]) were obtained from the Gene Expression Omnibus (GEO) using the Geoquery package (version 3.9) [56]. Of these 145 primary tumors, 110 were subtyped as classical and 35 as basal-like tumors. The association between different genes based on their normalized expression levels was analyzed using the linear regression method in R software (version 3.6.1). Visualization was accomplished using the ggplot2 package (version 3.3.2) [57].

### 4.12. Panc1-CCN1-KO Rescue

We reintroduced CCN1 expression into the Panc1-CCN1-KO line C7 using the AdEasy adenoviral vector system (Agilent Technologies, Santa Clara, CA, USA, #240009) and the virus AdE1-CMV-mCCN1 [58]. We infected the cells with 1 × 10^9^ viruses per 24-well plate for 4 h. After infection, the cells were washed three times with cell culture medium before proceeding with the cell culture process as previously described. The restored CCN1 expression was confirmed through Western blot analysis. As a control, we used the virus VQAd-CMV-eGFP at the same concentration on the same cell line, and the infection was confirmed through fluorescent imaging.

## Figures and Tables

**Figure 1 ijms-25-09369-f001:**
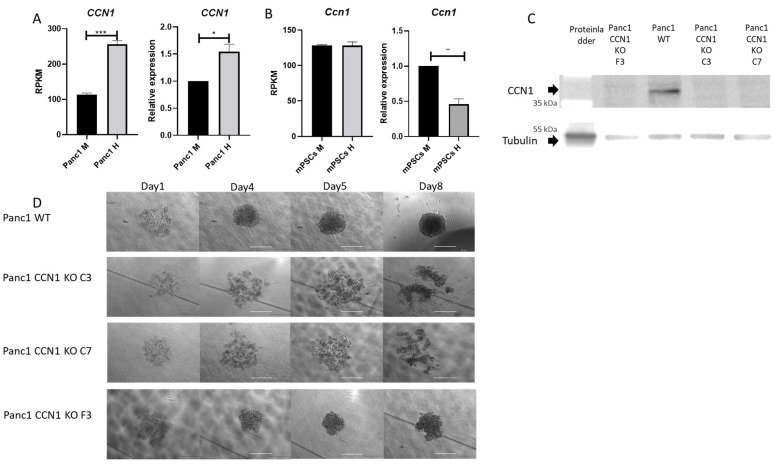
CCN1 expression in Panc1 is increased by co-culturing with mPSCs. (**A**,**B**) The expression level of CCN1/Ccn1 in Panc1-WT (**A**) and mPSCs (**B**) from monospheroids and heterospheroids on day 5 was analyzed via RNA-seq and RT-PCR. Represented are RPKM (Reads Per Kilobase per Million mapped reads) values from the published expression profiling [25], and the mRNA fold changes presented here were normalized to Panc1 or mPSCs monospheroids. All data shown are means with SEM from three independent biological replicates. (**C**) The representative Western blot illustrates the expression of CCN1 protein levels in Panc1-WT and 3 different Panc1-CCN1-KO clones (C3, C7, and F3) cultured in 2D. (**D**) Representative pictures of Panc1-WT and Panc1-CCN1-KO clones C3, C7, and F3 monospheroids. Scale bars represent 500 μm. M, monospheroid; H, heterospheroid. *, *p* < 0.05; **, *p* < 0.01; ***, *p* < 0.001.

**Figure 2 ijms-25-09369-f002:**
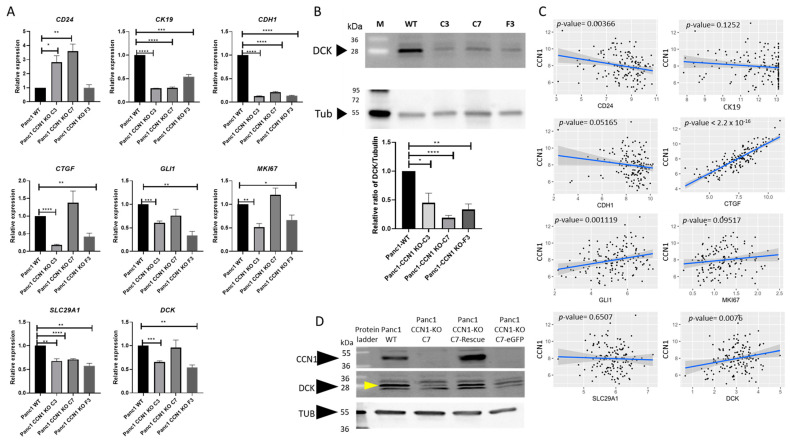
CCN1 negatively regulates stemness and positively affects gemcitabine transporter and activating genes. (**A**) The mRNA expression of the stemness marker gene CD24, the epithelial marker genes CK19 and CDH1, matricellular protein CTGF/CCN2, SHH signaling mediator GLI1, proliferation marker MKI67, gemcitabine transporter SLC29A1, and gemcitabine-activating enzyme DCK in Panc1-WT and Panc1-CCN1-KO clones (C3, C7, and F3) from monospheroids were analyzed with qRT-PCR. The mRNA fold changes were normalized to Panc1-WT monospheroids. All data shown are means with SEM from three independent biological replicates. (**B**) The protein level of DCK in Panc1-WT and Panc1-CCN1-KO clones (C3, C7, and F3) from 2D culture was analyzed with Western blot (WB). The relative ratio of DCK was quantified via DCK amount per lane/Tubulin amount per lane and then normalized to Panc1-WT. *, *p* < 0.05; **, *p* < 0.01; ***, *p* < 0.001; ****, *p* < 0.0001. n = 4, M = protein weight marker. (**C**) The associations between the expression of these marker genes and CCN1 were fitted using linear regression based on the microarray dataset GSE71729. n = 145 patient samples. (**D**) Protein levels of CCN1, DCK (yellow arrow), and Tubulin for Panc1 WT, Panc1-CCN1-KO C7, and the CCN1-rescued Panc1-CCN1-KO C7 carried out with adenoviral infection. The control for adenoviral infection was carried out with Panc1-CCN1-KO C7 cells infected with adenovirus containing an eGFP expression cassette.

**Figure 3 ijms-25-09369-f003:**
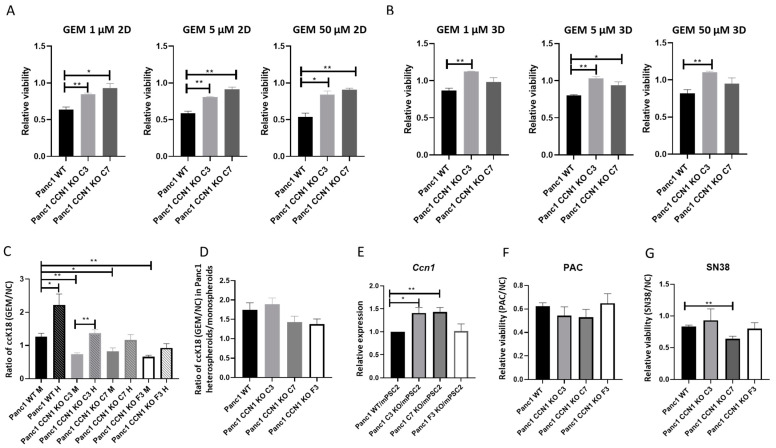
CCN1 increases chemotherapy sensitivity for gemcitabine but not paclitaxel and SN38. (**A**) The relative cell viability of Panc1-WT and Panc1-CCN1-KO (C3 and C7) cells grown in 2D culture after 72 h treatment with different doses of gemcitabine were measured via CellTiter-Glo^®^ 3D Cell Viability Assay and normalized to NC (negative control). (**B**) The relative cell viability of Panc1-WT and Panc1-CCN1-KO (C3 and C7) cells in monospheroids after 72 h treatment with different concentrations of gemcitabine were measured via CellTiter-Glo^®^ 3D Cell Viability Assay and normalized to NC. (**C**) The ratios of epithelial specific caspase-cleaved cytokeratin 18 (ccCK18) for Panc1-WT and Panc1-CCN1-KO (C3, C7, and F3) cells from monospheroids and heterospheroids after 72 h treatment with 50 µM gemcitabine were measured via M30 Apoptosense^®^ CK18 kit. (**D**) The relative ratios of the epithelial-specific cleaved ccCK18 following 72 h gemcitabine treatment of Panc1-WT and Panc1-CCN1-KO (C3, C7, and F3) cells from heterospheroid versus monospheroid culture are shown. (**E**) The mRNA expression of Ccn1 in mPSCs from heterospheroids co-cultured with either Panc1-WT or Panc1-CCN1-KO (C3, C7, and F3) cells is depicted. The mRNA fold changes were normalized to mPSCs co-cultured with Panc1 WT. (**F**) The relative cell viability of Panc1-WT and Panc1-CCN1-KO (C3, C7, and F3) cells from monospheroids after 72 h treatment with paclitaxel [1 μM] were measured via CellTiter-Glo^®^ 3D Cell Viability Assay and normalized to NC. (**G**) The relative cell viability of Panc1-WT and Panc1-CCN1-KO (C3, C7, and F3) cells from monospheroids after 72 h treatment with SN38 [1 μM] were measured via CellTiter-Glo^®^ 3D Cell Viability Assay and normalized to NC. All data shown are means with SEM of at least three independent biological replicates. *, *p* < 0.05; **, *p* < 0.01. M, monospheroid; H, heterospheroid; NC, negative control; GEM, gemcitabine; PAC, paclitaxel, SN38, active form of irinotecan.

**Figure 4 ijms-25-09369-f004:**
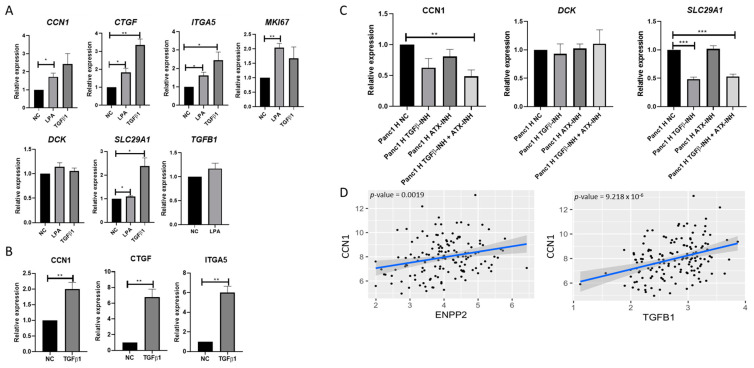
LPA and TGFB1 upregulate CCN1 expression in Panc1. (**A**) Analyses of mRNA expression of selected genes in Panc1-WT monospheroids cultured in low-serum condition following stimulation with 20 μM LPA or 5 μg/mL recombinant TGFB1 for 72 h. (**B**) Analyses of mRNA expression of selected genes in Panc1-WT monospheroids cultured in high-serum condition following stimulation with 5 μg/mL recombinant TGFB1 for 72 h. (**C**) The mRNA expressions of CCN1, DCK, and SCL29A1 in Panc1 cells from heterospheroids cultured in high-serum condition stimulated with 5 ng/mL TGFB receptor kinase inhibitor, or 10 μM ATX inhibitor PF8380, or a combination of both inhibitors for 72 h are depicted. The mRNA fold changes presented were normalized NC. All data shown are means with SEM of three biological replicates. *, *p* < 0.05; **, *p* < 0.01; ***, *p* < 0.001. (**D**) The associations between ENPP2, TGFB1, and CYR61(CCN1) were fitted using linear regression based on the microarray dataset GSE71729. n = 145 patient samples. ATX, autotaxin; INH, inhibitor.

**Figure 5 ijms-25-09369-f005:**
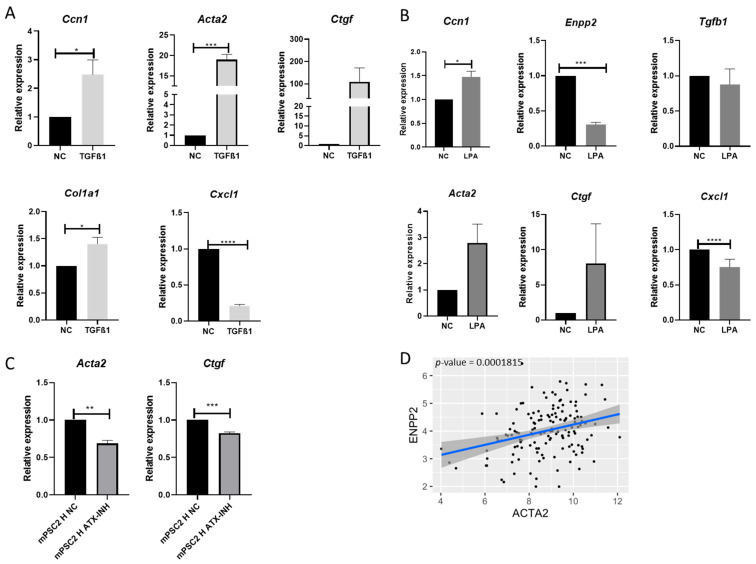
TGFB1 and LPA positively regulate Ccn1 in mPSCs and shift mPSCs to the myCAF subtype. (**A**) Analyses of mRNA expression of genes in mPSCs monospheroids stimulated with 5 μg/mL recombinant TGFB1 for 72 h. (**B**) Analyses of mRNA expression of genes in mPSCs monospheroids stimulated with 20 μM LPA for 72 h. The fold changes in the expression of selected genes were normalized to NC. (**C**) Analysis of the mRNA expression of genes in mPSCs from heterospheroid co-cultured with Panc1-WT cells stimulated with 10 μM ATX/ Enpp2 inhibitor PF8380 on day 5 for 72 h is depicted. The mRNA fold changes were normalized to NC. All data shown are means with SEM of three biological replicates. (**D**) The associations between ACTA2 and ENPP2 using linear regression based on the microarray dataset GSE71729. n = 145 patient samples. NC, negative control. *, *p* < 0.05; **, *p* < 0.01; ***, *p* < 0.001, ****, *p* < 0.0001.

## Data Availability

Raw RNA-seq data have been deposited in the NCBI Gene Expression Omnibus and are accessible through GEO Series accession numberGSE169539 [25].

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
