# Peer review of "The Crosstalk Analysis between mPSCs and Panc1 Cells Identifies CCN1 as a Positive Regulator of Gemcitabine Sensitivity in Pancreatic Cancer Cells"

_ijms, 2024, doi:10.3390/ijms25179369_

Round 1

Reviewer 1 Report

Comments and Suggestions for Authors

Dear Authors,

The authors set up a system of co-culture of the pancreatic cancer cell line Panc1 with mPSCs and, using Panc1 KO cells, showed that increased expression of CCN1 in Panc1 cells by mPSCs contributes to increased sensitivity to gemcitabine and prevents the undifferentiated state. Stimulation of TGF-β1 and LPA also increased the expression of CCN1 and genes involved in gemcitabine transport and metabolism. This is an interesting finding showing that stromal cells are involved in gemcitabine sensitivity and differentiation of pancreatic cancer cells, and I think that this finding is important in terms of pancreatic cancer therapy.

Regarding the Panc1 KO clone using CRISPR-Cas9, I think that a more accurate analysis of CCN1 KO might have been possible by creating a second clone using a different guide RNA, considering off-targets.

I would like the authors to answer the following three questions.

1. The most important findings are those from clinical specimens. While some gene expression results from clinical pancreatic cancer specimens support the authors' results in Panc1-CCN1-KO cells from both monospheroids (Fig. 2A) and heterospheroids (Supplementary Fig. 2), others differ; in Figure 2C, there does not appear to be a positive correlation with CCN1 for CK19, CDH1, and SLC29A1. What do the authors think about these? In addition, clinical information on microarray dataset GSE71729, such as whether the cases are early-stage pancreatic cancer or whether many of them have advanced stages, should be described.

2. Stimulation with LPA significantly increased the mRNA expression of CCN1 but not DCK. TGF-β1 stimulation also did not affect DCK expression. However, in Figures 2A and 2B, DCK mRNA expression in Panc1-WT and Panc1 CCN1 KO C7 does not change much, but there are significant differences in protein levels, suggesting that DCK mRNA expression does not correlate well with protein expression. In Figures 4A and 4C, the effects on TGF-β1 and LPA stimulation and their inhibitors do not change DCK mRNA expression, but does it make a difference at the protein level?

3. Is the regulation of CD24, CK19, CDH1, CTGF, GLI1, MKI67, SLC29A1, and DCK at the transcriptional level by CCN1 working directly on these promoter regions? In other words, is CCN1 a transcriptional regulator?

I would like to have the following Minor points corrected.

Minor points:

1. line 38: “may” is correct, not “my”.

2. Figure 1: Figure legend should have "*, P < 0.05; **, P < 0.01; ***, P < 0.001".

3. Figure 2C & 4D: The vertical axis of the figure should be CCN1, not CYR61. CCN1 is the official symbol.

4. Figure 3: In Figure legend, "***, P < 0.001" is not necessary since it is not seen in the figure.

5. Suppl Fig. 4A: What is mPSC2?

6. line 207-208: I think RNA-seq (data not shown) is correct, not RNA-seq (Supplementary Fig. 4C).

7. line 211: I think Supplementary Fig. 4C is correct, not Supplementary Fig. 4B, D. In fact, Supplementary Fig. 4D does not exist.

8. line 335: autotaxin/ENPP2 expression is not shown in Supplementary Fig. 4C. “data not shown” is correct.

9. line 362: What are myCAFs mPSCs?

Author Response

Response to Reviewer 1:

We are very grateful for the reviewer’s constructive criticism and suggestions. We believe that the manuscript has become significantly clearer thanks to the reviewer’s input. Please find our detailed point-by-point reply below. In the revised manuscript, changed text has been marked in red.

On top of the reviewers’ requests, we have introduced the following text/changes:

Line 112: The word “enzyme” has been replaced by “metabolizing”, which we believe fits better to the following text.

In Figure 1 we explained the abbreviation RPKM with “Reads Per Kilobase per Million mapped reads”.

Line 540-541: Acknowledgements: Part of this work was carried out at the Karolinska Genome Engineering Facility (KGE).

Comment 1. The most important findings are those from clinical specimens. While some gene expression results from clinical pancreatic cancer specimens support the authors' results in Panc1-CCN1-KO cells from both monospheroids (Fig. 2A) and heterospheroids (Supplementary Fig. 2), others differ; in Figure 2C, there does not appear to be a positive correlation with CCN1 for CK19, CDH1, and SLC29A1. What do the authors think about these? In addition, clinical information on microarray dataset GSE71729, such as whether the cases are early-stage pancreatic cancer or whether many of them have advanced stages, should be described.

Response 1: We fully agree with the reviewer that not all gene expression changes due to KO of  CCN1 in Panc1 cells are reflected 1:1 in patient expression data of figure 2C. Since there is a substantial genetic heterogeneity of cancer cells between PDAC patients and even within a patient, this is not unexpected and the reason for the recent development of personalized medicine underlined by the statement “one does not fit all”. In the best of cases, Panc1 cells do reflect one out of the many patients included in the microarray dataset GSE71729. In addition, since CCN1 is a hub/platform for multiple signaling pathways, different mutations affecting those pathways can have different net outcomes in different cells/clones/patients.

A more detailed information regarding the samples analysed in the microarray dataset GSE71729 has been added to Materials and Methods section “Microarray dataset analysis” lines 512-514.

Comment 2. Stimulation with LPA significantly increased the mRNA expression of CCN1 but not DCK. TGF-β1 stimulation also did not affect DCK expression.

Response 2. This is correct and we state this also in the discussion on lines 329-333.

Comment 2. However, in Figures 2A and 2B, DCK mRNA expression in Panc1-WT and Panc1 CCN1 KO C7 does not change much, but there are significant differences in protein levels, suggesting that DCK mRNA expression does not correlate well with protein expression.

Response 2: We have been puzzled by this observation as well. Since the experiments were repeated several times with the same result, we can only spectulate that the posttranscriptional regulation of DCK protein is influenced by other factors in this clone. Nevertheless, in comparison to “wt”-Panc1, C7 shows increased viability with a lower apoptosis rate after gemcitabine treatment, speaking for decreased DCK activity (Fig.3).

Comment 2. In Figures 4A and 4C, the effects on TGF-β1 and LPA stimulation and their inhibitors do not change DCK mRNA expression, but does it make a difference at the protein level?

Response 2: This is a very interesting question, which we will follow up in our future work in order to better understand the effects of TGFB/LPA stimulation on CCN1 expression and the genes involved in gemcitabine sensitivity.

Comment 3. Is the regulation of CD24, CK19, CDH1, CTGF, GLI1, MKI67, SLC29A1, and DCK at the transcriptional level by CCN1 working directly on these promoter regions? In other words, is CCN1 a transcriptional regulator?

Response 3. CCN1 is a secreted protein acting as a hub for different signaling pathways (esp. integrins) and other proteins such as CCN2/CTGF in the extracellular space. Since CCN1 has a secretory signal sequence it is very unlikely to act as a transcription factor and there is also  no hint in this direction in the literature. Regarding this, we have added some more information on CCN1 on lines 64 to 71.

Comment “Minor points”:

  1. line 38: “may” is correct, not “my”. We thank the reviewer for this correction.
  2. Figure 1: Figure legend should have "*, P < 0.05; **, P < 0.01; ***, P < 0.001". We agree with the reviewer.
  3. Figure 2C & 4D: The vertical axis of the figure should be CCN1, not CYR61. CCN1 is the official symbol. We thank the reviewer for this observation. This has been changed in the new version of the manuscript, but not highlighted in red letters/symbols.
  4. Figure 3: In Figure legend, "***, P < 0.001" is not necessary since it is not seen in the figure. The requested text has been removed.
  5. Suppl Fig. 4A: What is mPSC2? The “2” in mPSC2 is a typo and has been removed.
  6. line 207-208: I think RNA-seq (data not shown) is correct, not RNA-seq (Supplementary Fig. 4C). We apologize for not being absolutely clear. We did not detect any ENPP2 expression in Panc1 cells. We added this information now on line 212.
  7. line 211: I think Supplementary Fig. 4C is correct, not Supplementary Fig. 4B, D. In fact, Supplementary Fig. 4D does not exist. We have clarified this issue further in the text by introducing “of Tgfb1” and “mPSC” on line 215 as well as “Supplementary Fig. 4B, C" (line 217) showing the Tgfb1/TGFB1 mRNA expression in mPSCs and Panc1 cells, respectively.
  8. line 335: autotaxin/ENPP2 expression is not shown in Supplementary Fig. 4C. “data not shown” is correct. The correct information has been introduced on line 342.
  9. line 362: What are myCAFs mPSCs? We have removed “mPSCs” in line 369, which in our experiments develop a more myCAF-like signature due to the direct interaction with the Panc1 cancer cells.

Reviewer 2 Report

Comments and Suggestions for Authors

The authors have diligently looked at transcript expression of various genes to show increased expression of CCN1, and that the potential mechanism could be through LPA and TGFb, However, majority of the experiments are looking at transcript which does not always co-relate to protein expression and even function. The authors are missing key functional analysis throughout (expect of measuring viability in Figure 3) to show the importance of CCN1 and CAFs in gemcitabine resistance. The authors have a nice heterospheroid model that can used to address the functional aspect of the manuscript and increase its impact. Additionally, the manuscript is written in a way that all readers are in the field. Kindly provide proper rationales, so that people not in the field can also follow – as an example, in the introduction, authors explain what CCN1 is, however the rationale to look at CCN1 is missing; In Figure 2, the authors look at a lot of different genes from different pathways affected by CCN1, but why did the authors choose to focus on DCK for their validation studies?

Other points to improve the manuscript are as follows:

·       Fig 1B: Kindly explain the discrepancy in RNA-seq vs qPCR expression of Ccn1 in mPSCs M and mPSCs H.

·       Fig 1D, Fig 3E: Kindly explain the diff between clones F3 and C3/C7 when all have CCN1 KO.

·       Fig 2A: Comparing the 3 clones – the trends of gene expression are not similar, kindly explain the inconsistent results between the 3 clones and the trends do not match even if they are all CCN1 KO – kindly explain

·       Fig 2B: Based on gene expression of DCK in Fig 2A, there is no change in transcript of DCK in clone C7 compared to WT, however, by protein, C7 has the least protein expression!!

·       Fig 4A: the authors should check their data, specifically, CCN1 expression post TGFb stimulation. The authors say in their text that TGFb does not affect CCN1 expression, however, the mean of the bar seems to be more than LPA and NC, and yet LPA is significant, but not TGFb. Is there an outlier that has pulled the mean higher?

Author Response

Response to Reviewer 2:

We are very grateful for the reviewer’s constructive criticism and suggestions. We believe that the manuscript has become significantly clearer thanks to the reviewer’s input. Please find our detailed point-by-point reply below. In the revised manuscript, changed text has been marked in red.

On top of the reviewers’ requests, we have introduced the following text/changes:

Line 112: The word “enzyme” has been replaced by “metabolizing”, which we believe fits better to the following text.

In Figure 1 we explained the abbreviation RPKM with “Reads Per Kilobase per Million mapped reads”.

Line 540-541: Acknowledgements: Part of this work was carried out at the Karolinska Genome Engineering Facility (KGE).

The authors have diligently looked at transcript expression of various genes to show increased expression of CCN1, and that the potential mechanism could be through LPA and TGFb, However, majority of the experiments are looking at transcript which does not always co-relate to protein expression and even function. The authors are missing key functional analysis throughout (expect of measuring viability in Figure 3) to show the importance of CCN1 and CAFs in gemcitabine resistance. The authors have a nice heterospheroid model that can used to address the functional aspect of the manuscript and increase its impact. Additionally, the manuscript is written in a way that all readers are in the field. Kindly provide proper rationales, so that people not in the field can also follow – as an example, in the introduction, authors explain what CCN1 is, however the rationale to look at CCN1 is missing;

We thank the reviewer to bring up this important point. We have therefore introduced a new text with additional important information on CCN1 on lines 85 to 90.

In Figure 2, the authors look at a lot of different genes from different pathways affected by CCN1, but why did the authors choose to focus on DCK for their validation studies?

 We agree that the investigation of DCK is not really obvious to the reader at this point. To clarify this, we introduced the information that we had found these genes upregulated in Panc1 cells when cocultured with mPSCs in heterospheroids (Liu et al. 2021, Translational oncology 14, 101107-101107; Suppl. Fig. 7B, C) on lines 121 to 126.

Other points to improve the manuscript are as follows:

Fig 1B: Kindly explain the discrepancy in RNA-seq vs qPCR expression of Ccn1 in mPSCs M and mPSCs H.

Despite the fact that present RNA-seq methods and data analysis approaches are pretty robust, we felt that validation by qPCR would significantly improve the scientific validity of the data. Comparing the RNA-seq data with the qPCR data demonstrated that the results go almost always “in the same direction” (Fig. 1A; Suppl. Fig. 4A-C), even if the fold differences are not the same, which might be explained by the huge methodical difference between them. In the case of Ccn1, we even uncovered an exciting case of crosstalk between cancer and stromal cells using qPCR. Compared to mPCS monospheroids, Ccn1 in heterospheroids becomes downregulated in parallel with CCN1 upregulation in Panc1 cells. In combination with Panc1-CCN1-KO cells, Ccn1 becomes again upregulated, apparently compensating for the lack of CCN1.

Fig 1D, Fig 3E: Kindly explain the diff between clones F3 and C3/C7 when all have CCN1 KO.

This and the following two questions are dealing with the same problem. We have selected several CCN1-KO clones from single cells. Panc1 cells are aneuploid cells with an unstable genome like most cancer cells. It is therefore possible to select single cells that all have null mutations for CCN1, which we confirmed by sequencing, but with slightly different genomes and therefore with slightly different responses on certain research questions. We are, however, convinced of the conclusions we have drawn from our work, because they are based on repeated, independent experiments and there are always at least 2 out of the three KO-clones showing in the same direction.

Fig 2A: Comparing the 3 clones – the trends of gene expression are not similar, kindly explain the inconsistent results between the 3 clones and the trends do not match even if they are all CCN1 KO – kindly explain

Please see reply to previous comment.

Fig 2B: Based on gene expression of DCK in Fig 2A, there is no change in transcript of DCK in clone C7 compared to WT, however, by protein, C7 has the least protein expression!!

We have been puzzled by this observation as well. Since the experiments were repeated several times with the same result, we can only spectulate that the posttranscriptional regulation of DCK protein is influenced by other factors in this clone. Nevertheless, in comparison to “wt”-Panc1, C7 shows increased viability with a lower apoptosis rate after gemcitabine treatment, speaking for decreased DCK activity (Fig.3).

Fig 4A: the authors should check their data, specifically, CCN1 expression post TGFb stimulation. The authors say in their text that TGFb does not affect CCN1 expression, however, the mean of the bar seems to be more than LPA and NC, and yet LPA is significant, but not TGFb. Is there an outlier that has pulled the mean higher?

It has been described in the literature that TGFB stimulates CCN1 expression. In our experiments, this reached not statistical significance due to high variation, as the reviewer correctly assumes.

Round 2

Reviewer 2 Report

Comments and Suggestions for Authors

The authors have addressed majority of the comments, making the manuscript easier for the readers to follow. However, the authors are still missing functional validation to support their claim of the importance of CCN1 and CAFs in PDAC gemcitabine resistance. Majority of the data (except for measuring viability) are all transcriptional analysis, which does not always co-relate with function.

Author Response

Comment from the reviewer: The authors have addressed majority of the comments, making the manuscript easier for the readers to follow. However, the authors are still missing functional validation to support their claim of the importance of CCN1 and CAFs in PDAC gemcitabine resistance. Majority of the data (except for measuring viability) are all transcriptional analysis, which does not always co-relate with function.

Response:

Dear Reviewer 2

We do not really agree to have too few functional data on the mechanistic relationship between CCN1, PSCs/CAFs and gemcitabine sensitivity in Panc1 cells. We hope to rectify this impression by adding information not included in the original manuscript. Please consider the following facts:

We demonstrated opposite regulation of CCN1/Ccn1 in Panc1-, mPSC/CAF mono-and heterospheroids at the transcriptional level. Validation of our results on the protein level failed we did not find species-specific antibodies for unique identification of mouse Ccn1 from mPSCs/CAFs. We tried the mouse-specific antibodies MAB4864 and AF4055, both R&D Systems, without success. We always observed cross reactivity with human CCN1.

We show viability only for Panc1 wt and CCN1-KO cell (Fig. 3A, B; Suppl. Fig. 3A, B), because this assay cannot distinguish viability between mPSCs and Panc1 cells in heterospheroids.

However, using an epithelial/Panc1 cell-specific apoptosis assay, we demonstrated decreased gemcitabine sensitivity of the Panc1-CCN1-KO cells vs Panc1-wt cells in mono- and PSC/CAF-containing heterospheroids (Fig. 3C).

We show decreased mRNA expression for the gemcitabine transporter SLC29A1 and the key enzyme DCK which initiates the activation cascade of gemcitabine conversion into the toxic agent in Panc1-CCN1-KO cells vs Panc1-wt cells (Fig.2A).

We demonstrated a strongly reduced DCK protein expression in Panc1-CCN1-KO cells vs Panc1-wt cells (Fig. 2B), which can be rescued by adenoviral re-expression of CCN1 in Panc1-CCN1-KO cells (Fig. 2D), which is a strong indication for the mechanistic connection between CCN1 and DCK. We also tested the apoptosis rate of the CCN1-rescued Panc1-CCN1-KO cells vs Panc1 eGFP-Panc1-CCN1-KO cells, which showed generally increased apoptosis and slightly further increased apoptosis following treatment with 50 micromolar gemcitabine in the CCN1-rescued Panc1-CCN1-KO cells. We decided to not include these results (please see attached figure for the reviewer) into the manuscript, because the adenoviral infection by itself means a significant stress for the cell, increasing the apoptosis rate (ca. 8x higher in adenovirus infected vs non-infected cells) to an extent that is close to the maximal apoptosis rate of the cells, that is difficult to increase further by gemcitabine treatment.

In addition, we also tried several SLC29A1/hENT antibodies (PA5-142422, Invitrogen; LS-B12116, LS Bio), but unfortunately without success.

In summary, we are confident to have shown sufficient data to demonstrate a causal connection between CCN1 and gemicitabine sensitivity in the stromal/cancer cell containing 3D cell culture system used by us.

Round 3

Reviewer 2 Report

Comments and Suggestions for Authors

Thank you for addressing the reviewer comments.